# Risk of Liver Cirrhosis and Hepatocellular Carcinoma after Fontan Operation: A Need for Surveillance

**DOI:** 10.3390/cancers12071805

**Published:** 2020-07-06

**Authors:** Jun Sik Yoon, Dong Ho Lee, Eun Ju Cho, Mi Kyoung Song, Young Hun Choi, Gi Beom Kim, Yun Bin Lee, Jeong-Hoon Lee, Su Jong Yu, Haeryoung Kim, Yoon Jun Kim, Jung-Hwan Yoon, Eun Jung Bae

**Affiliations:** 1Department of Internal Medicine and Liver Research Institute, Seoul National University College of Medicine, Seoul 03080, Korea; yojusi@naver.com (J.S.Y.); yunbin@hanmail.net (Y.B.L.); pindra@empal.com (J.-H.L.); ydoctor2@hanmail.net (S.J.Y.); yoonjun@snu.ac.kr (Y.J.K.); yoonjh@snu.ac.kr (J.-H.Y.); 2Department of Internal Medicine, Busan Paik Hospital, Inje University College of Medicine, Busan 47392, Korea; 3Department of Radiology, Seoul National University College of Medicine, Seoul 03080, Korea; dhlee.rad@gmail.com (D.H.L.); choiyounghun@gmail.com (Y.H.C.); 4Division of Pediatric Cardiology, Department of Pediatrics, Seoul National University Children’s Hospital, Seoul National University College of Medicine, Seoul 03080, Korea; mksong52@hanmail.net (M.K.S.); ped9526@snu.ac.kr (G.B.K.); 5Department of Pathology, Seoul National University College of Medicine, Seoul 03080, Korea; medannabel@gmail.com

**Keywords:** Fontan-associated liver disease, hepatocellular carcinoma, liver cirrhosis, surveillance

## Abstract

Liver cirrhosis and hepatocellular carcinoma (HCC) are serious late complications that can occur after the Fontan procedure. This study aimed to investigate the cumulative incidence of cirrhosis and HCC and to identify specific features distinguishing HCC from benign arterial-phase hyperenhancing (APHE) nodules that developed after the Fontan operation. We retrospectively enrolled 313 post-Fontan patients who had been followed for more than 5 years and had undergone ultrasound or computed tomography (CT) of the liver between January 2000 and August 2018. Cirrhosis was diagnosed radiologically. The estimated cumulative incidence rates of cirrhosis at 5, 10, 20, and 30 years after the Fontan operation were 1.3%, 9.2%, 56.6%, and 97.9%, respectively. Multiphasic CT revealed that 18 patients had APHE nodules that were ≥1 cm in size and showed washout in the portal venous phase (PVP)/delayed phase, which met current noninvasive HCC diagnosis criteria. Among them, only seven patients (38.9%, 7/18) were diagnosed with HCC. After cirrhosis developed, the annual incidence of HCC was 1.04%. The appearance of washout in the PVP (*p* = 0.006), long time elapsed since the initial Fontan operation (*p* = 0.04), large nodule size (*p* = 0.03), and elevated serum α-fetoprotein (AFP) level (*p* < 0.001) were significantly associated with HCC. In conclusion, cirrhosis is a frequent late complication after Fontan operation, especially after 10 years, and HCC is not a rare complication after cirrhosis development. Diagnosis of HCC should not be based solely on the current imaging criteria, and washout on PVP and clinical features might be helpful to differentiate HCC nodules from benign APHE nodules.

## 1. Introduction

The Fontan operation is performed in patients with a functional or anatomic univentricular heart and connects systemic venous blood directly to the lungs without a pump. In the Fontan circulation, chronic venous hypertension, congestion of the systemic veins, and decreased cardiac output occur [1]. Many of the late post-Fontan complications, including Fontan-associated liver disease, are associated with these unique physiologic features.

Fontan-associated liver disease (FALD) includes a wide range of structural and functional disorders of the liver secondary to hemodynamic changes that follow the Fontan operation. It develops due to chronic liver congestion caused by the elevated systemic venous pressure and low cardiac output of the Fontan circulation. The spectrum of FALD ranges from mild hepatic congestion to cirrhosis; these may lead to the complications of portal hypertension and even hepatocellular carcinoma (HCC) [2]. Along with hepatic decompensation caused by long-lasting cirrhosis, HCC is one of the most serious late complications in patients after the Fontan operation. Therefore, surveillance for HCC should be warranted in post-Fontan patients [3,4,5]. However, guidelines about when to start screening for cirrhosis and HCC surveillance are still unclear due to limited information about the natural history of FALD [6].

Benign arterial-phase hyperenhancing (APHE) nodules are frequently observed after the Fontan operation [5,7,8,9,10,11,12]. These nodules histologically correspond to regenerative nodules or focal nodular hyperplasia [8,12]. The exact pathophysiology behind benign APHE nodules is unknown, though they are thought to be of vascular origin in view of their radiologic features and underlying Fontan physiology [13,14]. Benign APHE nodules might show washout on portal venous and/or delayed phase imaging [5,15]. In particular, when the washout is observed in nodules larger than 1 cm in size, it is consistent with current noninvasive imaging diagnostic criteria of HCC [16,17] and may be difficult to differentiate from HCC. Therefore, it is important to investigate the clinical and imaging features in order to differentiate benign APHE nodules from HCC.

Therefore, the aims of this study were (1) to assess the cumulative incidence rates of cirrhosis and HCC and (2) to identify specific features that can distinguish HCC from benign APHE nodules in patients after the Fontan operation.

## 2. Results

### 2.1. Patients

Among 332 patients, 313 patients had undergone radiologic exams of the liver between the study periods and were enrolled in the study. The 19 patients who had not undergone radiologic evaluation of liver during the study period had a relatively short time elapsed since the initial Fontan operation (median, 7.5 years) and did not show presumptive evidence of portal hypertension (e.g., median platelet count, 267,000/μL). The demographics and clinical characteristics of the 313 enrolled patients are summarized in Table 1. The median time elapsed from the date of the initial Fontan operation to the last image date was 18.6 years (interquartile range (IQR), 13.5–23.4).

### 2.2. Development of Cirrhosis

During the follow-up periods of 5882.7 person-years, 2012 imaging studies were examined in the 313 enrolled patients. The median number of imaging studies per person was 5 (IQR, 3–8) in the follow-up periods. Of these, abdominal ultrasound was performed 1065 times and abdominal CT scans or cardiac CT scans, which extended over the upper abdomen were performed 947 times. According to the aforementioned radiological diagnostic criteria, 221 of 313 patients (70.6%) were diagnosed with cirrhosis. Patients diagnosed with cirrhosis had significantly lower platelet count, greater spleen diameter, and higher noninvasive blood markers for fibrosis than those without cirrhosis (all *p* < 0.001; Appendix A). The ratios of atriopulmonary connection (APC) or lateral tunnel (LT) types in patients diagnosed with cirrhosis were significantly higher than those without cirrhosis (*p* < 0.001). The cumulative incidence rates of cirrhosis at 5, 10, 20, and 30 years after the Fontan operation were 1.3%, 9.2%, 56.6%, and 97.9%, respectively (Figure 1A). The cumulative incidence rate increased gradually until 10 years after the Fontan operation and then increased rapidly thereafter. We also calculated the cumulative incidence rate of the subgroup of 198 patients who were followed up within a 3-year interval. The cumulative incidence rates of cirrhosis at 5, 10, 20, and 30 years after the Fontan operation in this subgroup were 1%, 7.3%, 46.5, and 95.3%, respectively, which are similar to those of the entire population (Appendix A).

### 2.3. Clinical Characteristics Associated with Cirrhosis

We evaluated clinical characteristics associated with development of cirrhosis, including hemodynamic parameters measured by catheterization. Of 313 enrolled patients, 132 underwent catheterization at least more than once. At the time of the last catheterization, 53 patients had cirrhosis and 79 patients did not have cirrhosis (Appendix A). As expected, the patients diagnosed with cirrhosis had significantly longer time elapsed since the Fontan operation than those without cirrhosis (median, 18.6 vs. 10.6 years; *p* < 0.001). In addition, patients with cirrhosis had significantly more frequent presence of postoperative arrhythmia, lower arterial oxygen saturation, lower platelet counts, higher total bilirubin and gamma-glutamyl transferase (GGT) levels, and higher noninvasive blood markers for fibrosis than those without cirrhosis (all *p* < 0.05). The time elapsed since the initial Fontan operation (≥13.7 vs. <13.7 years; adjusted odds ratio (aOR): 4.15; 95% CI: 1.86–9.27; *p* = 0.001) and total bilirubin level (≥1.1 vs. <1.1 mg/dL; aOR: 2.62; 95% CI: 1.16–5.88; *p* = 0.02) were independently associated with the development of cirrhosis (Appendix A and Table 2).

### 2.4. Development of HCC

During the follow-up periods, 7 of 313 patients (2.2%) developed HCC and were diagnosed either by histology (*n* = 4) or clinical features (*n* = 3) (Appendix A). All three patients with clinically diagnosed HCC had elevated serum AFP levels at the time of detection (median, 141.6 ng/mL; range, 39.6–720.0), and these AFP levels declined to <20 ng/mL after the first treatment (median, 6.0 ng/mL; range, 4.1–9.7).

Death occurred in 24 of 313 patients during follow-up periods. Among them, two patients died of HCC progression. No patients underwent liver transplantation. The cumulative incidence rates of HCC at 10, 20, and 30 years after the Fontan operation were 0.3%, 0.8%, and 8.5%, respectively (Figure 1B), after considering competing risk of death. During a follow-up of 5875.9 person-years from the Fontan operation, seven patients developed HCC, with an annual incidence of 0.12%. However, after cirrhosis was diagnosed, the annual incidence of HCC was much greater: 1.04% (7 HCCs per 672.7 person-years).

### 2.5. Features Distinguishing HCC from Benign APHE Nodules

Eighty-two patients underwent multiphasic abdominal CT during the follow-up period to further evaluate hepatic nodules found in imaging (triple-phase CT scan: 54 patients; quadruple-phase CT scan: 28 patients). APHE nodules were observed in 42 patients and most of them (40/42) had radiologic findings of cirrhosis. Thirty patients had APHE nodules larger than 1 cm; they all had radiologic findings of cirrhosis. Among the 30 patients, washout was observed on both PVP and DP images in five patients and only on DP images in 13 patients. Based on imaging findings, assigned Li-RADs categories in these 30 patients were as follows: LR-3 in 12 patients, LR-4 in 9 patients, and LR-5 in 9 patients. There was no HCC in 12 patients with LR-3. One of nine patients with LR-4 had HCC (11.1%), and six of nine patients with LR-5 eventually had HCC (66.7%).

The imaging diagnostic performance in detecting HCC using different washout criteria for 30 patients with APHE nodule larger than 1 cm was summarized in Appendix A. When only PVP washout was considered to indicate HCC, the sensitivity, specificity, positive predictive value (PPV), and negative predictive value (NPV) were 71.4% (5/7), 100% (23/23), 100% (5/5), and 92.0% (23/25), respectively. However, if washout on the PVP and/or DP was considered to indicate HCC, which is the current imaging criterion for HCC diagnosis, the sensitivity, specificity, PPV, and NPV were 100% (7/7), 52.2% (12/23), 38.9% (7/18), and 100% (12/12), respectively.

Two of seven patients diagnosed with HCC underwent gadoxetic-acid-enhanced liver MRI, and the HCC had low signal intensity on the hepatobiliary phase image (Figure 2). Among the 11 patients diagnosed with benign nodules, 6 underwent gadoxetic-acid-enhanced liver MRI, and all APHE lesions had high signal intensity on the hepatobiliary phase image (Figure 3). With regards to serum tumor marker, six of the seven HCC patients had AFP levels ≥20 ng/mL, whereas all the 11 patients diagnosed with benign nodules had AFP levels <20 ng/mL. Those two parameters, washout on PVP (71.4% vs. 0%; *p* = 0.006) and serum AFP levels (median, 160.6 vs. 2.9 ng/mL; *p* < 0.001), showed significant differences between patients with HCC and those without (Table 3). Moreover, the time elapsed since the initial Fontan operation (median, 29.7 vs. 21.9 months; *p* = 0.04) and largest APHE nodule diameter (median, 4.0 vs. 1.7 cm; *p* = 0.03) were significantly greater in patients with HCC than those without.

## 3. Discussion

The aims of this study were to assess the cumulative incidence rates of cirrhosis and HCC and to identify specific features that can distinguish HCC from benign APHE nodule in patients after the Fontan operation. The primary findings of the study were that the cumulative incidence rate of cirrhosis increased rapidly 10 years after the Fontan operation and HCC was not uncommon in patients who had long-term follow-up after the Fontan operation, especially after development of cirrhosis. In addition, the positive predictive value of current noninvasive imaging diagnosis criteria for HCC was only 38.9% (7/18) in patients who underwent the Fontan operation. The presence of nodules that exhibited a washout appearance in the PVP, the long time elapsed since the initial Fontan operation, and a large diameter were helpful in differentiating between HCC and benign APHE nodules along with elevated serum AFP level.

We diagnosed cirrhosis using imaging studies, including ultrasound or CT, in this study. Although the gold standard for diagnosis of cirrhosis is liver biopsy, it is difficult to perform in post-Fontan patients due to the high risk of bleeding; this is because post-Fontan patients are known to have coagulopathies, and many are taking anticoagulants [18]. To date, the performance of other current clinical tools that estimate hepatic fibrosis has not been validated for the diagnosis of cirrhosis in post-Fontan patients [19]. Although diagnosis of cirrhosis using imaging studies has limitations, studies have reported that abnormal echotexture on an ultrasound is associated with biopsy-proven severe fibrosis in post-Fontan patients [2,20]. In addition, CT is widely used to evaluate cirrhosis in post-Fontan patients, and CT findings were reported to be correlated with histopathological findings [21]. In our study, the presence of cirrhosis on ultrasound or CT was associated with signs of portal hypertension, such as low platelet count, large spleen diameter, and noninvasive blood markers for hepatic fibrosis. Therefore, considering the results of aforementioned studies and ours, imaging in post-Fontan patients might be clinically valuable and provide potentially important information regarding the development of cirrhosis.

A recent retrospective study showed that cirrhosis had developed in 86% of post-Fontan patients by 20 years after the Fontan operation [21]. Similarly, the cumulative incidence rate of cirrhosis increased rapidly as the time after the Fontan operation exceeded 10 years, and the majority of patients had developed cirrhosis by 30 years after the Fontan operation in our study. APC or LT types were more frequently performed procedures in patients with cirrhosis than those without cirrhosis. This might be explained by the significant transition in the Fontan procedures over time, from an APC type to an LT type, and from LT type to an extracardiac connection type. In addition, the type of Fontan operation was not an independent predictor of cirrhosis in the multivariable analysis, which indicates that the exposure time to Fontan circulation is a more important predictor of cirrhosis. Indeed, time elapsed since the Fontan operation was significantly associated with cirrhosis in the multivariable analysis, which was consistent with previous studies [21,22]. The longer the time elapsed after the Fontan operation, the longer the exposure to hepatic congestion, which may lead to cirrhosis.

In our study, the PPV of current noninvasive imaging criteria consisting of APHE followed by washout on PVP and/or DP was only 38.9%. Even in LR-5 category, the PPV was only 66.7%. Considering that the theoretical PPV of the LR-5 category should be almost 100%, this result clearly demonstrated the limitation of applying current noninvasive imaging diagnosis criteria for HCC in post-Fontan patients. Among the various imaging findings, washout in the PVP was significantly associated with the presence of HCC, and this result was well correlated with a previous study conducted by Wellis et al. [5]. In addition, both high specificity and PPV could be obtained when washout on PVP only was used to diagnose HCC. In contrast, washout in the DP was not specific for HCC in our post-Fontan patients. Owing to long-standing hepatic congestion, hepatic parenchymal abnormalities including fibrosis, hepatic sinusoidal dilatation, and increased extracellular space could develop in the post-Fontan liver and could increase contrast retention on the DP [15,23,24,25]. Therefore, washout in the DP in post-Fontan patients may not indicate a change in the blood supply of the hepatocellular nodule itself, but may reflect retained contrast material in the background normal liver parenchyma [5]. Our results might support the recommendation of the Liver Imaging Reporting and Data System [26] that imaging diagnosis criteria for HCC should not be applied to patients with cardiac cirrhosis, such as post-Fontan patients.

Our study had some limitations. Firstly, the majority of patients were diagnosed with cirrhosis based on radiologic findings without histologic confirmation. Instead, we diagnosed cirrhosis with radiologic findings according to strict criteria. Presence of portal-hypertension-related complications and increased fibrosis markers in these patients supported our diagnostic reliability of cirrhosis. Secondly, three patients with HCC were diagnosed by clinical features without histologic confirmation. However, the patients had elevated AFP level at the time of nodule detection, and their serum AFP levels decreased immediately after TACE to the normal range. Moreover, all the patients showed compact lipiodol uptakes after TACE. Given the elevated tumor marker and response to TACE, we believe that the possibility of benign liver tumors in these three patients was quite low, almost negligible.

## 4. Materials and Methods

### 4.1. Study Population

A total of 332 patients who had undergone the Fontan operation and were followed at Seoul National University Hospital (Seoul, Republic of Korea) for more than 5 years were considered for this retrospective cohort study. Among them, patients who had undergone at least one abdominal ultrasound or computed tomography (CT) scan of the liver between January 2000 and August 2018 were included in the study. Patients had undergone initial screening for FALD 5–10 years after the Fontan operation and were subsequently followed depending on clinical need. Laboratory tests including complete blood count, liver functions, and serology of viral hepatitis were examined at the time of the Fontan operation. Follow-up examinations of complete blood count and liver function tests were performed every 3–6 months after the Fontan operation, and imaging tests were considered in cases of any suspicious findings of liver disease progression such as thrombocytopenia or hyperbilirubinemia. The individualized follow-up intervals and imaging modalities were determined by the cooperation of a multidisciplinary team including pediatric cardiologists, radiologists, and hepatologists. For patients diagnosed with cirrhosis, the date of initial diagnosis of cirrhosis was further investigated by reviewing their records before 2000. The results of cardiac catheterization were also recorded when available. Because the surveillance interval was not standardized at the time of evaluation, we further selected a subgroup of patients who were followed up by abdominal imaging and laboratory tests every 1 to 3 years according to the recent statement from the American Heart Association [27]. This study was conducted in accordance with the 1975 Declaration of Helsinki and approved by the Institutional Review Board of Seoul National University Hospital (IRB No. 1812-158-998).

### 4.2. Evaluation of Cirrhosis

Cirrhosis was assessed based on radiologic findings referring to clinical data. Radiologic assessments for cirrhosis included ultrasound and CT scan, and an independent experienced board-certificated abdominal radiologist (D.H.L. with a 10-year experience in liver imaging) reviewed all images. Cirrhosis was suspected when imaging studies including ultrasound and/or CT showed liver surface nodularity, as the presence of liver surface nodularity has been reported to be the most common and sensitive imaging feature of cirrhosis [28,29,30]. However, it is recognized that liver surface nodularity in imaging study of post-Fontan patients may not be associated with liver dysfunction [19]. Therefore, in order to provide a more specific diagnosis of cirrhosis in this study, the presence of cirrhosis was determined by the concurrent presence of liver surface nodularity and signs of portal hypertension, including presence of splenomegaly (defined as size greater than 12 cm in the largest bipolar diameter at the splenic hilum), presence of ascites, and/or presence of portosystemic collateral vessels (i.e., varices), in imaging study [2,31,32,33].

### 4.3. Predictors of Cirrhosis

In patients who underwent cardiac catheterization, clinical and hemodynamic parameters associated with development of cirrhosis were evaluated. The presence of cirrhosis was investigated through a radiologic exam performed within 1 year from the date of the catheterization, and laboratory data within 1 month after catheterization were recorded. Noninvasive blood markers for hepatic fibrosis, including Model for End-Stage Liver Disease excluding INR (MELD-XI), Forns index, aspartate transaminase to platelet ratio index (APRI), and Fibrosis-4 index (FIB-4), were also calculated. The values of noninvasive imaging modalities for liver fibrosis such as transient elastography or acoustic radiation force impulse elastography could not be investigated because these modalities were not routinely used as screening tools for cirrhosis in our study population. The history of warfarin, diuretics, and angiotensin-converting enzyme inhibitors or angiotensin-receptor blockers was recorded when the patients had taken any of these medications for longer than 1 year.

### 4.4. Diagnosis of APHE Nodules

For patients who underwent a triple- or quadruple-phase abdominal CT scan during the follow-up period, the presence of APHE nodules was recorded. Firstly, all APHE nodules with the potential for HCC were identified in the liver on late arterial phase (LAP) images, and the size and number of APHE nodules were determined. Thereafter, washout characteristics of APHE nodules were evaluated on both portal venous phase (PVP) and delayed phase (DP) images. Gadoxetic-acid-enhanced liver magnetic resonance imaging (MRI) was performed in some of patients with APHE nodules for further characterization, and the nodule signal intensity on hepatobiliary phase images was evaluated when available.

### 4.5. Diagnosis of HCC

Diagnosis of HCC was made by either histologic examination or clinical diagnosis in consideration of imaging features of nodules and serum alpha-fetoprotein (AFP) levels [34]. Typical imaging features for HCC were defined as an APHE nodule ≥1 cm in size with washout in the PVP/DP according to the current noninvasive imaging diagnosis criteria for HCC [16,17]. Presence of a nodule showing the typical HCC pattern in combination with elevated AFP levels (≥20 ng/mL) at the time of detection was considered as HCC. In patients with typical nodules and AFP levels <20 ng/mL, we performed follow-up imaging using CT or MRI at 3–6 months interval because many APHE nodules in post-Fontan livers could be benign nodules [5,35]. When the APHE nodule size increased in follow-up imaging, liver biopsy was considered for pathological confirmation.

### 4.6. Statistical Analysis

Logistic regression analysis was used to identify independent predictors of being diagnosed with cirrhosis. The cumulative incidence rate of cirrhosis/HCC was calculated using the Kaplan–Meier method: person-years were calculated from the date of the Fontan operation until the date when cirrhosis/HCC was diagnosed or the date of the last image, taking into account competing risk of death that occurred free of HCC. The data cutoff date was 30 June 2019. Two annual incidences of HCC were also calculated: from the date of the Fontan operation and from the date of diagnosis of cirrhosis. *p* values <0.05 were considered significant. Statistical analyses were performed using R software, Version 3.5.2 (R Foundation for Statistical Computing, Vienna, Austria).

## 5. Conclusions

The cumulative incidence rate of cirrhosis at 10 years after the Fontan operation was only 9.2%, but it increased rapidly thereafter. Thus, we suggest that screening for cirrhosis be started 10 years after the Fontan operation, even if the patients do not have any signs or symptoms of hepatic dysfunction. The occurrence of HCC was not uncommon, especially in post-Fontan patients with cirrhosis; thus, surveillance for HCC should be considered in that population. The application of current noninvasive imaging diagnosis criteria for HCC has limitations in post-Fontan patients, and washout in the PVP of APHE nodules, long time elapsed since the initial Fontan operation, and large nodule diameter might be helpful for diagnosis of HCC together with elevated serum AFP level.

## Figures and Tables

**Figure 1 cancers-12-01805-f001:**
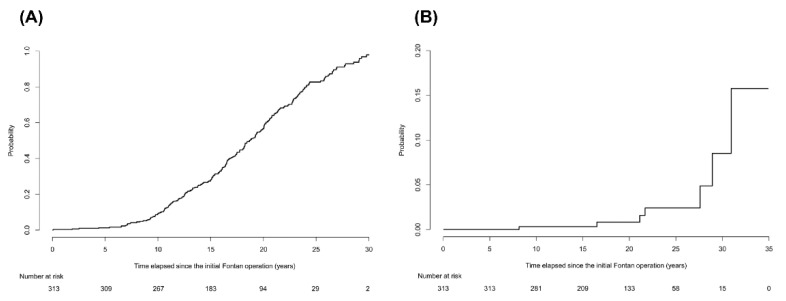
The cumulative incidence rates of cirrhosis and hepatocellular carcinoma in patients after the Fontan operation. (**A**) The cumulative incidence rates of cirrhosis at 5, 10, 20, and 30 years after the Fontan operation were 1.3%, 9.2%, 56.6%, and 97.9%, respectively. The cumulative incidence rate increased gradually until 10 years after the Fontan operation and then increased rapidly thereafter. (**B**) The cumulative incidence rates of HCC at 10, 20, and 30 years after the Fontan operation were 0.3%, 0.8%, and 8.5%, respectively, after considering competing risk of death.

**Figure 2 cancers-12-01805-f002:**
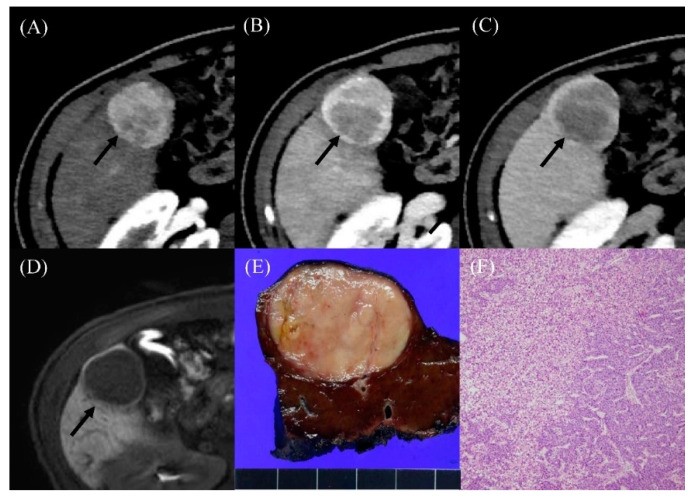
Radiologic findings in a 36-year-old male who developed hepatocellular carcinoma 15 years after the Fontan operation. (**A**) Contrast-enhanced arterial-phase axial CT image shows a 4-cm-sized enhancing mass in segment V of the liver (arrow). (**B**) The posterior portion of mass shows washout in the portal venous phase axial CT image (arrow). (**C**) The washout of the mass was more evident on the delayed phase image (arrow). (**D**) On hepatobiliary phase of gadoxetic-acid-enhanced liver MR, the mass shows low signal intensity (arrow). The serum AFP level was elevated to 274.4 ng/mL. (**E**) The resected specimen revealed a well-circumscribed, solid, pinkish mass. (**F**) Histological examination showed well-differentiated hepatocellular carcinoma with a trabecular pattern (H and E, ×40).

**Figure 3 cancers-12-01805-f003:**
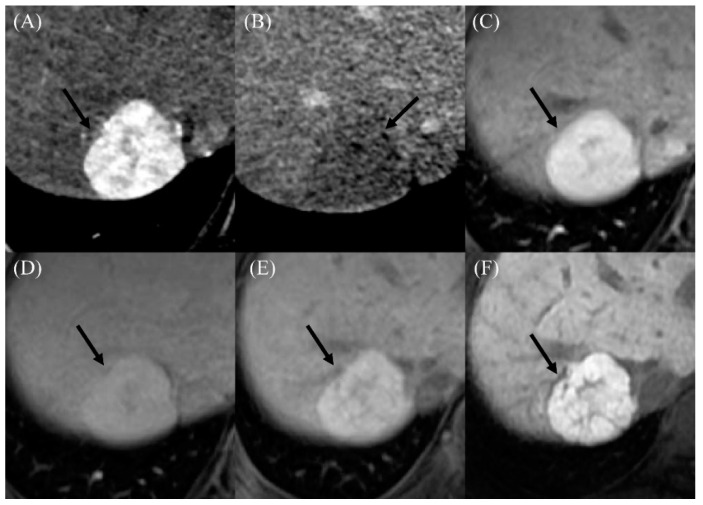
Imaging findings in a 22-year-old male who developed a benign arterial-phase enhancing nodule 18 years after the Fontan operation. (**A**) The contrast-enhanced arterial-phase axial CT image shows a 3-cm-sized enhancing mass in segment VII of the liver (arrow). (**B**) This mass exhibits the appearance of washout in the delayed phase image (arrow). (**C**) The gadoxetic-acid-enhanced arterial-phase axial MR image obtained 1 month after the CT scan also shows a 3-cm-sized enhancing mass in segment VII of the liver. (**D**) This mass has high signal intensity in the transitional phase image (arrow). (**E**) In the hepatobiliary phase image, this mass has high signal intensity (arrow), suggesting the presence of functioning hepatocytes. The serum AFP level was 4.7 ng/mL in this patient. (**F**) In the follow-up hepatobiliary phase of the gadoxetic-acid-enhanced MR image obtained 2 years after initial detection, the size and signal intensity of the mass were unchanged.

**Table 1 cancers-12-01805-t001:** Demographics and clinical characteristics of the enrolled patients.

	Total Population (*n* = 313)
Male sex	204 (65.2%)
Age at the initial Fontan operation (years)	2.7 (2.1–3.9)
Type of Fontan operation	
Atriopulmonary connection	54 (17.3%)
Lateral tunnel	126 (40.3%)
Extracardiac	133 (42.5%)
Time elapsed since the initial Fontan operation (years)	18.6 (13.5–23.4)
Age at the last follow-up date (years)	21.3 (16.8–27.6)
Hepatitis B virus infection	5 (1.6%)
Hepatitis C virus infection	2 (0.7%)
Development of cirrhosis	221 (70.6%)
Development of hepatocellular carcinoma	7 (2.2%)
Postoperative arrhythmia	87 (27.8%)
Protein-losing enteropathy	22 (7.0%)

Data are expressed as median (interquartile range) or n (%).

**Table 2 cancers-12-01805-t002:** Predictors of cirrhosis in patients after the Fontan operation.

	Univariable Analysis		Multivariable Analysis	
	Odd Ratio (95% CI)	*p* Value	Odd Ratio (95% CI)	*p* Value
Age at the initial Fontan operation (≥2.6 vs. <2.6 years)	1.56 (0.77–3.15)	0.21	-	-
Sex (male vs. female)	1.53 (0.71–3.29)	0.28	-	-
Type of Fontan operation		0.24	-	-
Atriopulmonary connection	1.00 (reference)			
Lateral tunnel	0.46 (0.18–1.13)			
Extracardiac	0.57 (0.22–1.48)			
Time elapsed since the initial Fontan operation (≥13.7 vs. <13.7 years)	5.37 (2.49–11.56)	<0.001	4.15 (1.86–9.27)	0.001
Fontan pressure (≥13 vs. <13 mmHg)	1.21 (0.60–2.46)	0.59	-	-
VEDP (≥9 vs. <9 mmHg)	1.37 (0.68–2.76)	0.38	-	-
SaO2 (≥92.4 vs. <92.4%)	0.62 (0.31–1.26)	0.19	-	-
Warfarin (yes vs. no)	1.96 (0.96–4.01)	0.06	-	-
Diuretics (yes vs. no)	1.76 (0.85–3.63)	0.13	-	-
Postoperative arrhythmia (yes vs. no)	2.42 (1.14–5.12)	0.02	-	0.20
Protein-losing enteropathy (yes vs. no)	0.56 (0.17–1.90)	0.35	-	-
Viral hepatitis (yes vs. no)	0.36 (0.04–3.32)	0.37	-	-
ACEI/ARB (yes vs. no)	0.88 (0.40–1.96)	0.76	-	-
Platelet count (≥191,000 vs. <191,000/μL)	0.38 (0.19–0.78)	0.01	-	0.49
Total bilirubin (≥1.1 vs. <1.1 mg/dL)	3.72 (1.76–7.86)	<0.001	2.62 (1.16–5.88)	0.02
AST (≥27.0 vs. <27.0 U/L)	0.61 (0.30–1.23)	0.16	-	-
ALT (≥20.0 vs. <20.0 U/L)	1.56 (0.77–3.16)	0.21	-	-
GGT (≥60.0 vs. <60.0 U/L)	2.57 (1.25–5.30)	0.01	-	0.06
Albumin (≥4.4 vs. <4.4 g/dL)	1.61 (0.79–3.30)	0.19	-	-
MELD-XI (≥6.82 vs. <6.82)	2.86 (1.38–5.89)	<0.001	-	0.36
Forns index (≥3.05 vs. <3.05)	5.07 (2.36–10.91)	<0.001	-	0.38
APRI (≥0.38 vs. <0.38)	1.87 (0.92–3.79)	0.08	-	-
FIB-4 (≥0.12 vs. <0.12)	3.50 (1.61–7.64)	<0.001	-	0.77

Data are expressed as median (interquartile range) or *n* (%). Continuous variables were categorized into two groups according to their median values. VEDP, ventricular end diastolic pressure; SaO_2,_ arterial oxygen saturation; ACEI, angiotensin-converting enzyme inhibitor; ARB, angiotensin II receptor blocker; AST, aspartate aminotransferase; ALT, alanine aminotransferase; CI, confidence interval; GGT, gamma-glutamyl transferase; MELD-XI, Model for End-Stage Liver Disease excluding INR; APRI, aspartate transaminase to platelet ratio index; FIB-4, Fibrosis-4 index.

**Table 3 cancers-12-01805-t003:** Comparison of clinical characteristics between patients diagnosed with HCC and benign APHE nodules after the Fontan operation.

	Patients Diagnosed with HCC (*n* = 7)	Patients Diagnosed with Benign APHE Nodules (*n* = 11)	*p* Value
Male sex	4 (57.1%)	4 (36.4%)	0.71
Type of Fontan operation			0.06
Atriopulmonary connection	5 (71.4%)	2 (18.2%)	
Lateral tunnel	2 (28.6%)	6 (54.5%)	
Extracardiac	0 (0.0%)	3 (27.3%)	
Time elapsed since the initial Fontan operation (years)	29.7 (24.8–30.4)	21.9 (16.7–23.3)	0.04
Number of APHE nodules	1.0 (1.0–6.0)	5.0 (1.5–9.0)	0.32
Largest APHE nodule diameter (cm)	4.0 (2.3–4.4)	1.7 (1.2–2.1)	0.03
Washout on PVP			0.006
No or unknown	2 (28.6%)	11 (100.0%)	
Yes	5 (71.4%)	0 (0.0%)	
Platelet count (×10^3^/μL)	122.0 (94.0–165.5)	191.0 (168.0–237.5)	0.08
Total bilirubin (mg/dL)	1.5 (0.9–2.8)	1.8 (1.3–2.7)	0.53
AST (U/L)	30.0 (23.0–34.0)	22.0 (21.0–24.5)	0.15
ALT (U/L)	25.0 (19.0–32.0)	16.0 (15.0–17.0)	0.08
GGT (U/L)	119.0 (60.0–123.0)	74.0 (60.5–85.5)	0.36
Albumin (g/dL)	4.2 (4.1–4.6)	4.5 (4.2–4.7)	0.52
AFP (ng/mL)	160.6 (90.6–483.7)	2.9 (2.3–3.5)	<0.001

Data are expressed as median (interquartile range) or *n* (%). HCC, hepatocellular carcinoma; APHE, arterial-phase hyperenhancing; PVP, portal venous phase; AST, aspartate aminotransferase; ALT, alanine aminotransferase; GGT, gamma-glutamyl transferase; AFP, alpha-fetoprotein.

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
