# Peer review of "Risk of Liver Cirrhosis and Hepatocellular Carcinoma after Fontan Operation: A Need for Surveillance"

_cancers, 2020, doi:10.3390/cancers12071805_

Round 1
Reviewer 1 Report
I read with attention the manuscript entitled: "Risk of Liver Cirrhosis and Hepatocellular Carcinoma after Fontan Operation: A Need for Surveillance" written by Yoon et al. Authors present a retrospective analysis of 313 post-Fontan patients who had been followed for more than 5 years. The radiological evaluation was a primary including criteria. The risk of developing cirrhosis and HCC increased over time. This result is very interesting, considering the high incidence reported by the authors. Near to 100% after 30 years, and 10% at 10 years. The risk of HCC on cirrhosis increase as well.
The manuscript is well presented, analysis is very clear. The discussion is well structured and comprehensive.
In my opinion, the manuscript deserves publication
Author Response
We revised the manuscript according to the reviewer's comment. The point by point response was attached as a file.

Reviewer 2 Report
A very well written paper dealing with an important issue for patients with this rare condition. Thank you for that.
There are only minor remarks:
First in supplementary table 1 there is a significant difference between patients with and without cirrhosis regarding the type of Fontan -Operation. This is not commented. Has the type of operation an influence on developement of cirrhosis or is this a result of different types used a different time points in history. This should be clarified.
Second; The cirrhosis is diagnosed by CT-criteria. Is there any correlation with non invasive markers like Fibroscan, elastography of serum scores? Do you have data on flow in portal vein?
Author Response

(The authors gave the same response as above.)
